# Spinal Meningioma Surgery in Octogenarians: Functional Outcomes and Complications over a 2-Year Follow-Up Period

**DOI:** 10.3390/medicina58101481

**Published:** 2022-10-18

**Authors:** Pavlina Lenga, Gelo Gülec, Awais Akbar Bajwa, Mohammed Issa, Karl Kiening, Basem Ishak, Andreas W. Unterberg

**Affiliations:** Department of Neurosurgery, Heidelberg University Hospital, 69120 Heidelberg, Germany

**Keywords:** spinal meningioma, spinal decompression, aging, risk factors

## Abstract

*Background and Objectives*: Population aging in industrial nations has led to an increased prevalence of benign spinal tumors, such as spinal meningiomas (SMs), in the elderly. The leading symptom of SM is local pain, and the diagnosis is confirmed after acute neurological decline. However, little is known about the optimal treatment for this frail patient group. Therefore, this study sought to assess the clinical outcome, morbidity, and mortality of octogenarians with SMs and progressive neurological decline undergoing surgery and to determine potential risk factors for complications. *Materials and Methods*: Electronic medical records dated between September 2005 and December 2020 from a single institution were retrieved. Data on patient demographics, neurological conditions, functional status, degree of disability, surgical characteristics, complications, hospital course, and 90-day mortality were collected. *Results*: Thirty patients aged ≥80 years who were diagnosed with SMs underwent posterior decompression via laminectomy and microsurgical tumor resection. The patients presented with a poor baseline history (mean CCI 8.9 ± 1.6 points). Almost all SMs were located in the thoracic spine (*n* = 25; 83.3%). Progressive preoperative neurological decline was observed in 21/30 (*n* = 21; 70%) patients with McCormick Scores (mMCS) ≥3, and their mean motor score (MS) was 85.9 ± 12.3. in the in-hospital and 90-day mortality rates were 6.7% and 10.0%, respectively. The MS (93.6 ± 8.3) and mMCS (1.8 ± 0.9) improved significantly postoperatively (*p* < 0.05). The unique risk factor for complications was the severity of comorbidities. *Conclusions*: Decompressive laminectomy and tumor removal in octogenarians with progressive neurological decline improved patient functional outcomes at discharge. Surgery seems to be the “state of the art” treatment for symptomatic SMs in elderly patients, even those with poor preoperative clinical and neurologic conditions, whenever there is an acceptable risk from an anesthesiological point of view.

## 1. Introduction

Spinal meningiomas (SMs) are mostly benign, slow-growing, intradural, extramedullary tumors and constitute 25–46% of all primary spinal tumors and approximately 12% of all meningiomas [1,2]. These lesions arise from the meningothelial cells of the arachnoid and are most prevalent in the thoracic spine [2,3,4].

SMs are slow-growing lesions that lead to chronic spinal canal compression and myelopathy. The leading symptom is local pain, although in a considerable number of patients, the diagnosis is confirmed after the occurrence of an acute neurological decline [5]. Several studies have suggested that the peak incidence of SMs occurs between the sixth and eighth decades [6,7]. Therefore, due to increasing life expectancy and accelerating improvements in medical care, the treatment of symptomatic SMs in the geriatric population has gained increasing attention. The primary goal of spinal tumor surgery is sufficient decompression via simple laminectomy or hemilaminectomy of the spinal cord and/or spinal roots by radical tumor resection to prevent worsening of the neurological status and recover preoperative neurological deficits. However, due to their poor baseline reserve, older patients are burdened by a higher risk of peri-and postoperative complications, which can result in poor surgical outcomes [8,9].

A few studies on older patients with SMs showed that advanced age might not be a contraindication for performing surgery and might support patients’ functional recovery [8,9,10,11,12]. Notwithstanding, there is a void in the literature of studies focusing specifically on the surgical treatment of octogenarians with SMs, who are touted to be at an increased risk of morbidity and mortality.

Considering the lack of robust clinical evidence, our study aimed to describe the clinical course of SMs in octogenarians with acute onset of neurological illness, assess morbidity and mortality rates, and determine potential risk factors for postoperative complications.

## 2. Materials and Methods

### 2.1. Study Design and Data Collection

We retrospectively evaluated the clinical and imaging data collected from our institution’s database between September 2005 and December 2020. This study was approved by the local ethics committee of our institution (approval no. 880/2021) and was conducted in accordance with the Declaration of Helsinki. The requirement for informed consent was waived owing to the retrospective nature of the study. Patients aged ≥80 years with histologically confirmed SMs across the thoracic and lumbar spine and acute-onset neurological decline were consecutively enrolled (Figure 1). No patient had an already radiologically or histologically diagnosed spinal meningioma; thus, a disease progression or a progressive spinal cord compression on MRI due to the tumor mass could not be evaluated. Therefore, we did not include a cross-over. Spinal deformity, traumatic or degenerative changes were evaluated using computed tomography (CT) and magnetic resonance imaging (MRI). The exclusion criteria were as follows: age <80 years, concurrent intracranial or cervical pathology, complete loss of disc height, and bony deconstruction resulting in kyphosis or subluxation of the vertebral column, vertebral collapse of >50%, bone necrosis, spinal deformity, and traumatic or degenerative changes. This patient subset was excluded from the current study to avoid biases in the functional outcomes due to the need for different surgical techniques, e.g., spinal instrumentation and laminectomy for tumor resection. Such factors might have led to a heterogenous cohort, which would have significantly altered the functional outcomes and even the mortality rates due to extension of surgery.

### 2.2. Patient Characteristics

Electronic medical records were assessed to obtain patient demographics, comorbidities, American Society of Anesthesiologists (ASA) scores, duration of surgery, number of treated spinal levels, peri- and postoperative complications, hospital length of stay (LOS), intensive care unit (ICU) stay, readmission, reoperation, and mortality. Comorbidities present before surgery were assessed using the age-adjusted Charlson comorbidity index (CCI) [13,14]. The CCI was calculated for each patient and classified as no comorbidity (CCI = 0), minimal comorbidity (CCI = 1 or 2), moderate comorbidity (CCI = 3–5), or severe comorbidity (CCI > 5). Pre-treatment neurological condition was assessed using the motor score (MS) of the American Spinal Injury Association (ASIA) impairment grading system (MS = 0, no muscle strength; MS = 100, healthy). Moreover, we adopted the modified McCormick Score (mMCS), an overall functional neurological classification scheme (grade I = normal gait, II = mild gait disturbance not requiring support, III = gait with support, IV = assistance required, and V = wheelchair needed) [15]. Post-treatment MS and mMCS scores were obtained from the last documented clinical encounter to assess functional outcomes. The extent of resection was defined using Simpson’s classification system [16]. All the patients presenting with neurological decline underwent posterior microsurgical decompression via laminectomy. Depending on the rate and degree of neurological compromise, operations were performed in an emergency or elective manner. Decision making was guided by presenting neurological status (measured by MS), concomitant underlying pathologies, the extent of the pathology, the prognosis of the disease, and the discretion of an experienced treatment team of neurosurgeons, neuroradiologists, and anesthesiologists. The attending spinal surgeon made the final decision. Routine clinical and radiological follow-up examinations were performed before discharge and 3, 6, 12, and 24 months after surgery. MRI of the spinal cord was performed at every outpatient visit to evaluate for tumor recurrence.

### 2.3. Statistical Analysis

Categorical variables are presented as numbers and percentages. Continuous variables are presented as mean and standard deviation, and the Shapiro–Wilk test was used to verify whether their distribution was normal. Baseline characteristics, duration of surgery, number of treated spinal levels, perioperative and postoperative complications, LOS, ICU stay, readmissions, reoperations, and mortality were compared groupwise using independent *t*-tests for continuous variables and chi-squared tests for categorical variables. The Wilcoxon rank test was used to evaluate changes in neurological status (measured by MS) and functional outcomes (measured by mMCS) at discharge. In the second-stage analysis, binary logistic regression was performed to identify risk factors for the occurrence of complications. Statistical significance was set at a *p*-value ≤ 0.05.

## 3. Results

### 3.1. Patient Demographic and Baseline Data

Over a period of 15 years, 30 patients aged ≥80 years who were diagnosed with SMs underwent posterior decompression via laminectomy and microsurgical tumor resection. The mean age was 82.6 ± 1.2 years with an overt predominance of the female gender (*n* = 19, 63.3%). The mean age-adjusted CCI was 8.9 ± 1.6 points, indicating a poor baseline reserve. Arterial hypertension, coronary heart disease, and myocardial infarction were the most prevalent comorbidities (*n* = 24, 80.0%; *n* = 19, 63.3%; and *n* = 15, 50.0%, respectively). Almost all SMs were in the thoracic spine (*n* = 25; 83.3%). Progressive neurological decline was observed in 21/30 (*n* = 21; 70.0%) patients with mMCS ≥3 and a mean MS of 85.9 ± 12.3, indicating the presence of a new motor deficit. A detailed breakdown of patient characteristics is presented in Table 1.

### 3.2. Surgical Characteristics and Clinical Scores

The mean surgical duration was 190.3 ± 67.4 min, with a mean blood loss of 433.3 ± 36.1 mL (Table 2). The mean number of decompressed levels was 1.5 ± 0.7. The mean ICU stay was less than one day, while the mean hospital stay lasted 11.2 ± 1.4 days. During hospitalization, two patients (6.7%) died due to severe respiratory decline caused by pneumonia. The mortality at 90 days was 10%, and the cause of death was not associated with spinal pathology in any patient. Three patients were readmitted because of myocardial infarction. Simpson grade 2 resection was conducted in 25/30 patients (83.3%); 90% of the SMs were World Health Organization (WHO) meningioma classification grade I and 10% were WHO grade II. No further surgeries were required during the follow-up period, and no tumor recurrence was seen on the follow-up MRI (mean follow-up, 26.3 ± 5.3 months). Most importantly, the MS and mMCS improved significantly after surgery. The postsurgery mean mMCS was 1.8 ± 0.9, indicating low levels of patient disability. Secondary instability was not present in any case, as evaluated by follow-up imaging.

### 3.3. Complications

The most prevalent complications were urinary tract infections (*n* = 5, 16.7%) and acute heart failure (*n* = 5, 16.7%) (Table 3). Cerebrospinal fluid (CSF) leakage occurred in one case, and a lumbar drain was placed for 5 days. After drain removal, the wound remained dry; hence, no revision surgery was required. Pulmonary embolism occurred in three patients (10%). Detailed data regarding the postoperative complications are shown in Table 4. In the second-stage analysis, potential risk factors for the occurrence of complications were examined (Table 5). Interestingly, patients’ comorbidities as measured by CCI were associated with complications, while the extent or duration of surgery, blood loss, and neurological deficits were not.

## 4. Discussion

With increasing life expectancy worldwide, the proportion of older patients, especially octogenarians with spinal tumors and meningiomas, is steeply increasing [6,7,17]. SMs might remain neurologically silent for longer intervals, causing unspecific subtle symptoms such as local or radicular pain. Moreover, these symptoms may be misinterpreted due to the poor baseline reserve of the elderly, thereby leading to substantial delays in their prompt diagnosis and subsequent management [17]. As the diagnosis is confirmed by the presence of progressive neurological deficits, it still poses a challenge to treat this subset of patients optimally with respect to the higher morbidity and mortality risks associated with their existing underlying diseases.

### 4.1. Summary of Findings

To the best of our knowledge, this is the first systematic analysis examining the clinical course of surgical treatment for SMs exclusively in patients aged ≥80 years. We assessed mortality and morbidity rates and determined potential risk factors for complications in octogenarians undergoing posterior surgical decompression and tumor resection due to progressive neurological decline. We found that octogenarians presented with high comorbidity rates (mean age-adjusted CCI, 8.9), indicating high grades of frailty. Notably, approximately half of the patients presented with higher grades of disability (mMCS ≥ 4), and all had at least one motor deficit (mean MS 85.9). The in-hospital and 90-day mortality rates were relatively high, at 6.7% and 10%, respectively. Post-surgery, the levels of dependency as defined by the mMCS and patients’ neurological deficits improved significantly with a mean mMCS of 1.8, indicating restoration of walking ability. Interestingly, higher rates of comorbidities were a unique risk factor for the occurrence of complications, while the duration or extent of surgery or patients’ neurological conditions were not. Almost all SMs were WHO I°, and Simpson Grade 2 resection was achieved in 83.3% of the patients. Over a two-year follow-up period, no further surgery due to secondary instability or tumor recurrence was necessary.

### 4.2. Review of Literature

While various surgeries, such as hip surgery, transurethral prostate resection, heart valve placement, and coronary bypass surgery, are performed routinely in the elderly, spinal surgery is still viewed with trepidation for the same patient cohort [18,19]. In the present study, the severity of comorbidities was high, raising concerns regarding the benefits of a surgical procedure, even in the presence of new neurological deficits. In a retrospective study of 30 older patients undergoing surgery for the resection of SMs, Morandi et al., reported recovery rates of 72% with a mean Solero score of 2, indicating low rates of disability, even in patients with a preoperative ASA score of III [11]. In line with these findings, Sacko et al., in their retrospective analysis of 102 patients over 70 years with SMs, stated that advanced age was not a contraindication, even in patients with severe comorbid illness, for performing surgery since the morbidity and mortality rates did not increase and the patients’ quality of life improved significantly [9]. In another case report of a 101-year-old female patient with progressive weakness of the lower extremities due to a tumor mass effect in SM, surgical resection resulted in good clinical outcomes with complete recovery of ambulation. It is important to note that the patient’s medical history was not significant except for mild congestive heart failure, which was fully compensated by medication [20]. In conjunction with the above-mentioned studies, Engel et al., set as a threshold an ASA score less than 3 for the elderly who opted for surgery for SM as an indicator for a higher rate of self-dependency postoperatively [21]. It is worth noting that patients aged ≥80 years were underrepresented; hence, the effectiveness of surgery is still controversial. However, since a chance for recovery seems to be present, we believe that surgery should be considered even in this frail cohort. A meticulous study of such patients and the potential benefits and shortcomings of surgical procedures should be discussed with the patients’ families and the patients themselves.

The present study reported that the diagnosis of SMs was initiated because of progressive neurological decline. Notably, about half of the enrolled patients initially presented with severe motor or sensory deficits (mMCS > 3) and concomitant high grades of dependency [22]. Consistent with these findings, Sandalcioglo et al., conducted a retrospective analysis of 131 individuals with SMs with a mean age of 69 years (range 17–88 years) and found that the vast majority of the cases (84%) had at least one motor or sensory deficit and 47% were unable to walk at the time of presentation [22]. The authors suggested that neurological deterioration was mainly observed in older patients (76–88 years) [22]. Nevertheless, postoperatively, only 12% of the patients were unable to walk, rates that were comparable with those mentioned in our study (mMCS = 3 in 16.7%). It is important to highlight that no explicit analysis has been conducted for outcomes in older patients. Similarly, in another SM study based on claim data, patients aged 70 years and older were significantly burdened by higher degrees of disability than occurred in younger patients (49.6% vs. 30.5; *p* < 0.001) [23]. However, surgical resection of SMs resulted in substantial amelioration of patients’ functional status, although older patients require more time for recovery, which is apparently due to their clinical vignette [23]. Older age at a given level of surgery and higher rates of underlying conditions were significant risk factors for less favorable outcomes. In line with these data, Sacko et al., in their retrospective study with a special focus on older patients (>70 years) with SMs, found that 84.3% of the cases suffered from bladder or bowel dysfunction, while 32.4% presented with spinal cord compression symptoms [9]. Reassuringly, surgery led to neurological improvement in 85.3% of patients, with 48.0% experiencing complete recovery [9].

In the clinical realm, neurosurgeons are routinely confronted with an increasing number of older patients. It is generally accepted that operative procedures in the elderly are accompanied by higher rates of complications or even death [18]. Notwithstanding, in cases of neurological deterioration (for example, due to SMs) surgery seems ostensibly a critical pillar for neurological recovery. Chronological age should not be a decisive factor. Previous evidence in a cohort of 102 older patients undergoing surgery for SMs suggested relatively low 90-day mortality rates of 2% [9], although a steep increase in deaths was seen one-year post-surgery. It should be noted that these deaths were unrelated to surgery. Similar findings were reported by Setzer et al., in 80 patients with a mean age of 61.9 years (range 20–91 years) [24]. The study group reported in-hospital mortality rates of 2% due to pulmonary embolism [24]. In concert with the above-mentioned studies, Morandi et al. observed no operative mortality in patients aged ≥70 years after microsurgical resection of SMs [11]. Likewise, Champeaux-Depond et al. found a very low mortality rate of 0.6% in a large cohort of 277 patients with SMs [23]. In contrast, we found higher in-hospital and 90-day mortality rates of 6.7% and 10%, respectively. These contradictory results may be attributable to the fact that only octogenarians with severe clinical profiles were included. The observed deaths were not related to surgery but were caused by severe respiratory failure and myocardial infarction. In light of these findings, comorbidities predispose patients to increased risks. For example, previous evidence suggests that CCI and ASA scores might be key tools for predicting postoperative complications in spinal surgery [25]. Therefore, a meticulous study of these debilitated patients and a clear discussion of both the benefits and drawbacks is mandatory, even in the setting of emergent surgery.

Increasing age is associated with higher morbidity and mortality in patients since this patient subset is debilitating due to its poor baseline history [26]; thus, spine surgeons may be reluctant to perform surgical procedures due to the potential peri- and postoperative complications. Hohenberger et al., retrospectively analyzed 45 patients with a mean age of 63 years with SM, reporting that 80% of patients undergoing surgery recovered fully after 12 months, while age did not substantially impact patients’ functional outcomes [27]. In another study, Capo et al. analyzed 72 patients aged 75 years and older and found increasing age and preoperative neurological status as significant prognostic factors for outcomes; however, age did not seem to be a contraindication for surgery [10]. Comorbidities and ASA scores significantly affected surgical outcomes. In conjunction with these findings, Sacko et al., advocated that age should not be considered a contraindication for surgery especially in patients with preoperative motor deficits, since surgery led to complete neurological recovery in 91.2% of the cases, and that most importantly, no death occurred, even in their cohort of patients aged 70 years and older [9]. However, it seems somehow surprising, but in the case of spinal meningioma, young age (<50 years) may indicate a risk factor for further surgery due to tumor recurrence [27,28,29]. That phenomenon might be attributable to the slow disease progression as well as to longer life expectancy in the younger age group. A concise overview of the related studies is provided in Table 6.

It is well known that older patients have a higher risk of peri- and postoperative complications. According to our findings, urinary tract infections, acute heart failure, and pulmonary embolism were the most prevalent adverse events after surgery. Interestingly, the CSF leakage rates were very low without the need for revision surgery. Most importantly, the severity of comorbidities was a unique risk factor for complications, while surgical characteristics, such as duration of surgery, the surgery itself, or even blood loss, were not. In line with these findings, Sacko et al., reported similar postoperative complications in elderly patients, with urinary tract infections being the most frequent [9]. However, separate risk analyses to identify potential risk factors have not been conducted. Kwee et al. also showed similar complication rates (overall prevalence of 21.7%) in their retrospective analysis of SMs [27]. Herein, the single patient with CSF leakage received a lumbar drain. In another review of literature on outcomes after SM surgery, expected complications were pulmonary embolism, pneumonia, stroke and myocardial infarction, while the incidence of CSF leakage is reported to be low (range 0–4%), as also shown in the present study [30]. Whereas the above-mentioned studies consisted of patients with a wide range of ages, the complications were akin to ours. Since our cohort included only patients aged ≥80 years, we observed a relatively long hospital stay of 11.2 days. This substantially longer hospitalization may be explained by institutional policy, as a longer hospital stay was mandatory to assure the best medical care and avoid clinical worsening in such a frail cohort.

The rationale for minimally invasive approaches is to preserve structures that may be important for spinal stability [31]. In a recent study on the optimal surgical approach for the resection of intradural spinal tumors, Goodarzi et al. did not find any significant differences between laminectomy and hemilaminectomy, and none of the patients from either group needed revision surgery due to secondary instability over a 6-year follow-up period [32]. In our series, laminectomy was performed for tumor resection, with no spinal instability resulting from the surgery. Experience with microsurgical techniques applied to disc surgery has shown that limited laminectomy is effective for both visualization and dural closure. Several studies have provided technical details for enhancing visualization through narrow corridors [31,32,33]. Another advantage of the posterolateral approach is that the tumor is first visualized, facilitating its dissection from the spinal roots and cord. Visualization of anteriorly located tumors is easy, with no or minimal cord rotation or manipulation.

### 4.3. Limitations

The strength of the current study is that it is the first to examine the clinical course and outcomes of octogenarians with SMs undergoing surgical resection. However, this study has some limitations. First, a relatively small cohort of patients was included. A systematic analysis focusing on this subset of patients is still lacking. Despite this, we feel that our findings produce a real-world picture of the disease course and management. Second, we were unable to present data on further adjuvant treatments, because patients with signs of instability were excluded from the study. Since we included only patients with acute-onset of neurological deficits and novel radiologically or histologically confirmed SM, a selection bias might have occurred. However, we provided data on a homogenous cohort; thus, a cross-over was not present which might have substantially altered the functional outcomes of the present study. Furthermore, a selection bias might have occurred solely due to the retrospective study design. A comparative analysis with younger patients may provide additional information concerning differences between outcomes and recurrence rates. Nevertheless, this study aimed to accentuate the uniqueness of the management of octogenarians with SMs since this cohort typically requires a multitude of health care interventions, especially in terms of surgical procedures. Moreover, octogenarians have only been incidentally included in previous studies due to their frailty, so we feel that this study showcases the need for surgery even in such a debilitating cohort and may help physicians in decision-making with respect to the emergency due to the acute neurological decline.

A separate analysis of such a group should be conducted because another therapy might be required in these patients. Indeed, large prospective studies are warranted to investigate therapeutic trends and clinical outcomes in this patient population.

## 5. Conclusions

Over the past several decades, patients ≥80 years have been considered “high-risk” candidates for spinal surgery due to the increased risk of perioperative morbidity and mortality. The present study supports the concept that the advanced age of patients might not be a contraindication for SM surgery since their quality of life can be improved substantially. However, the surgical treatment and its related complications, including potential morbidity, should be discussed with patients and their relatives. Nevertheless, quality of life can be preserved or improved in octogenarians with prompt assessment and early surgery after acute onset.

## Figures and Tables

**Figure 1 medicina-58-01481-f001:**
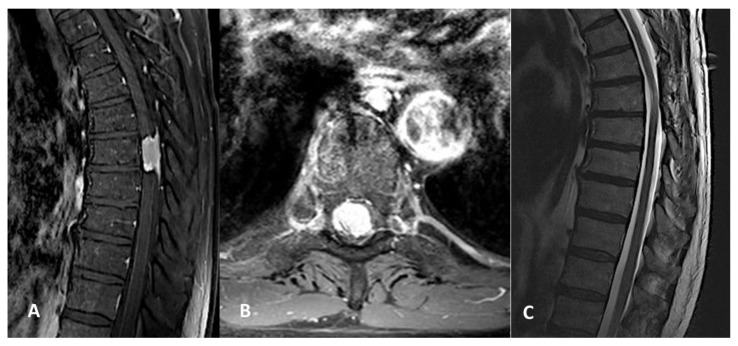
(**A**) Preoperative sagittal magnetic resonance image (MRI) showing a right thoracic spinal meningioma (SM) at level Th7. (**B**) Preoperative axial MRI showing the same SM. (**C**) Postoperative MRI demonstrating complete removal of the lesion.

**Table 1 medicina-58-01481-t001:** Baseline Characteristics.

Characteristic	Value
Number of patients	30
Age, years (mean, SD)	82.6 (1.2)
Sex (*n*, %)	
Male	11 (36.7)
Female	19 (63.3)
Body mass index, kg/m^2^ (mean, SD)	25.6 (4.3)
Comorbidities	
Age-adjusted CCI score (mean, SD)	8.9 (1.6)
Arterial hypertension (*n*, %)	24 (80.0)
Myocardial infarction (*n*, %)	15 (50.0)
Coronary heart disease (*n*, %)	19 (63.3)
Atrial fibrillation (*n*, %)	14 (46.7)
Heart failure (*n*, %)	9 (30.0)
COPD (*n*, %)	6 (20.0)
Diabetes mellitus Type II (*n*, %)	6 (20.0)
Renal failure (*n*, %)	9 (30.0)
Liver disease (*n*, %)	7 (23.3)
Gastrointestinal ulcer (*n*, %)	5 (16.7)
TIA/stroke (*n*, %)	8 (26.7)
Malignancy (*n*, %)	3 (10.0)
Dementia (*n*, %)	8 (26.7)
Previous spinal surgery (*n*, %)	1 (3.3)
ASA class (*n*, %)	
II	10 (33.3)
III	20 (66.7)
Location (*n*, %)	
Thoracic	25 (83.3)
Lumbar	5 (16.7)
Preoperative MS (mean, SD)	85.9 (12.3)
Modified McCormick Score (mMCS; *n*, %)	
mMCS 1	6 (20.0)
mMCS 2	3 (10.0)
mMCS 3	7 (23.3)
mMCS 4	8 (26.7)
mMCS 5	6 (20.0)
Histology (*n*, %)	
WHO I°	27.0 (90.0)
WHO II°	3 (10.0)

ASA, American Society of Anesthesiologists; CCI, Charlson comorbidity index; COPD, chronic obstructive pulmonary disease; MS, motor score of the American Spinal Injury Association grading system; mMCS, modified McCormick Score; SD, standard deviation; TIA, transient ischemic attack; WHO, World Health Organization meningioma classification grade.

**Table 2 medicina-58-01481-t002:** Peri- and postoperative surgical characteristics and clinical course of the 30 patients who underwent decompression surgery.

Characteristic	Value
Surgical duration, minutes	190.3 (67.4)
Number of levels decompressed	1.5 (0.7)
Estimated blood loss, mL	433.3 (36.1)
Hospital stay, days	11.2 (1.4)
ICU stay, days	0.5 (0.2)
Mortality	
In-hospital (*n*, %)	2 (6.7)
90-day (*n*, %)	3 (10.0)
30-day readmission (*n*, %)	3 (10.0)
MS	93.6 (8.3)
Modified McCormick Score (mMCS; *n*, %)	
mMCS 1	13 (43.3)
mMCS 2	12 (40.0)
mMCS 3	3 (10.0)
mMCS 4	2 (6.7)
mMCS 5	0 (0.0)
Simpson Grade (*n*, %)	
1	0 (0.0)
2	25 (83.3)
3	5 (16.7)
4	0 (0.0)
5	0 (0.0)

Except where otherwise indicated, values are mean (SD). ICU, intensive care unit; MS, motor score of the American Spinal Injury Association grading system; mMCS, modified McCormick Score.

**Table 3 medicina-58-01481-t003:** Occurrence of adverse events in the octogenarian patients (*n* = 30) who underwent decompression surgery.

Event	Number of Patients (%)
CSF leakage	1 (3.3)
Pulmonary embolism	3 (10.0)
Acute heart failure	5 (16.7)
Acute renal failure	3 (10.0)
Pneumonia	4 (13.3)
Urinary tract infection	5 (16.7)

CSF, cerebrospinal fluid.

**Table 4 medicina-58-01481-t004:** Comparison of baseline (before surgery) and discharge neurological condition and functional status scores.

	Decompression Baseline (*n* = 30)	Decompression Discharge (*n* = 30)	*p*-Value
MS	85.9 (12.3)	93.6 (8.3)	**<0.001**
mMCS	3.2 (1.4)	1.8 (0.9)	**<0.001**

All data are presented as mean (SD). MS, motor score of the American Spinal Injury Association grading system; mMCS, modified McCormick Score. The *p*-values presented in the bolded font indicate statistically significant results.

**Table 5 medicina-58-01481-t005:** Risk factors associated with the occurrence of complications.

Risk Factor	OR (% 95 CI)	*p*-Value
Age-adjusted CCI score	2.1 (1.1–4.3)	**0.004**
Preoperative MS	1.1 (0.9–1.2)	0.841
Preoperative mMCS	1.3 (0.5–3.6)	0.624
Duration of surgery	1.0 (0.9–1.1)	0.999
Estimated blood loss	1.1 (0.9–1.2)	0.996
Number of levels decompressed	3.7 (1.5–4.5)	0.226
Length of hospital stay	1.4 (1.2–3.2)	0.444

CCI, Charlson Comorbidity Index; CI, confidence interval; MS, motor score of the American Spinal Injury Association grading system; mMCS, modified McCormick Score; OR, odds ratio. The *p*-values presented in the bolded font indicate statistically significant results.

**Table 6 medicina-58-01481-t006:** Overview of recent studies on spinal meningioma with a special focus on patients’ outcomes with respect to the patient’s age.

	*n*	Age, y (Mean)	Functional Outcome (% of Improvement)	Recurrence Rates	Reoperation
(Nakamura et al., 2012) [29]	68	56	Not applicable	35.0% (age < 50 y)	17.6%
(Cohen-Gadol et al., 2003) [28]	40	34.5	82.0%	22.5%	22.5%
(Sacko et al., 2009) [9]	102	74.6	91.2	1.0%	0.0%
(Kwee et al., 2020) [27]	166	66	73.0%	7.2%	12.0%
(Hohenberger et al., 2020) [26]	45	63	80.0%	None	0.0%
(Capo et al., 2022) [10]	72	75	65.3%	None	None

## Data Availability

The data presented in this study are available on request from the corresponding author.

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
