# Peer review of "Spinal Meningioma Surgery in Octogenarians: Functional Outcomes and Complications over a 2-Year Follow-Up Period"

_medicina, 2022, doi:10.3390/medicina58101481_

Round 1

Reviewer 1 Report

The authors present a case series of elderly patients surgically treated for spinal meningiomas. In our practice is a common believe that elderly patients represent a high risk group and sometimes decisions of do not operate are made based only in age, and as population ages, surgical decisions over this group of patients need to be made based on evidence and functional status. This type of publications helps us to make correct decisions.

We make the following considerations to this fine work:

  1. Abstract: Results need to be optimised pointing only relevant findings.
  2. Methods: 
    1. we suggest to specify the cross over, retrospective nature of the study, and to clarify that is a case series.  
    2. We suggest to make a comparative analysis with the younger group of patients treated at your institution to prove if there is any difference in risks, surgical resultas and complications; that will highly improve the significance of your results. 
  3. Discussion:
    1. We suggest to summarize results from other papers in one table.

Author Response

A1. Abstract: Results need to be optimised pointing only relevant findings.

Comment

We thank the reviewer for this suggestion. The results in the abstract have been optimized to highlight the relevant findings.

Thirty patients aged ≥80 years who were diagnosed with SMs underwent posterior decompression via laminectomy and microsurgical tumor resection. The patients presented with a poor baseline history (mean CCI 8.9±1.6 points). Almost all SMs were located in the thoracic spine (n=25; 83.3%). Progressive preoperative neurological decline was observed in 21/30 (n=21; 70%) patients with McCormick Scores (mMCS) ³3, and their mean motor score (MS) was 85.9±12.3. in the in-hospital and 90-day mortality rates were 6.7% and 10.0%, respectively. The MS (93.6±8.3) and mMCS (1.8±0.9) improved significantly postoperatively (p<0.05). The unique risk factor for complications was the severity of comorbidities.

A2. Methods: 

A2.1. We suggest to specify the cross over, retrospective nature of the study, and to clarify that is a case series.  

Comment

We agree that the methods require some clarification.

Therefore, we have amended the “Methods” section as follows: (Also please refer to our response to Reviewer 2’s report, comment A1 if further information is required)

Patients aged ³80 years with histologically confirmed SMs across the thoracic and lumbar spine and acute-onset neurological decline were consecutively enrolled. No patient had an already radiologically or histologically diagnosed spinal meningioma; thus, a disease progression or a progressive spinal cord compression on MRI due to the tumor mass could not be evaluated. Therefore, we did not include a cross-over.

We have also revised the “Limitation” section as follows:

Since we included only patients with acute-onset of neurological deficits and novel radiologically or histologically confirmed SM, a selection bias might have occurred. However, we provided data on a homogenous cohort; thus, a cross-over was not present which might have substantially altered the functional outcomes of the present study. Furthermore, a selection bias might have occurred solely due to the retrospective study design. A comparative analysis with younger patients may provide additional information concerning differences between outcomes and recurrence rates. Nevertheless, this study aimed to accentuate the uniqueness of the management of octogenarians with SMs since this cohort typically requires a multitude of health care interventions, especially in terms of surgical procedures. Moreover, octogenarians have only been incidentally included in previous studies due to their frailty, so we feel that this study showcases the need for surgery even in such a debilitating cohort and may help physicians in decision-making with respect to the emergency due to the acute neurological decline.

A2.2. We suggest to make a comparative analysis with the younger group of patients treated at your institution to prove if there is any difference in risks, surgical results and complications; that will highly improve the significance of your results. 

Comment

However, the focus of the current study was to elucidate the outcomes exclusively for octogenarians with acute-onset neurological decline upon undergoing tumor resection via laminectomy. We think that a comparative analysis with a younger cohort would alter our study’s take-home message. Our primary hypothesis was to evaluate this unique subset of patients, which warrants thorough preoperative evaluation of their frailty attributable to their poor baseline reserve. We strongly believe that your suggested analysis would yield interesting results, but this can be analyzed in another comparative study focusing on clinical outcomes of different age brackets. However, if the reviewers and the editor further advise the addition of such an analysis, we will be glad to include it an additional revision of this manuscript. In the future, we aim to conduct a study determining the potential differences between outcomes in respect to the patient’s age.

We have discussed this point in our “Discussion” section as follows:

Increasing age is associated with higher morbidity and mortality in patients since this patient subset is debilitating due to its poor baseline history (Audisio et al., 2005); thus, spine surgeons may be reluctant to perform surgical procedures due to the potential peri- and postoperative complications. Hohenberger et al. retrospectively analyzed 45 patients with a mean age of 63 years with SM, reporting that 80% of patients undergoing surgery recovered fully after 12 months, while age did not substantially impact patients’ functional outcome (Hohenberger et al., 2020). In another study, Capo et al. analyzed 72 patients aged 75 years and older and found increasing age and preoperative neurological status as significant prognostic factors for outcomes; however, age did not seem to be a contraindication for surgery (Capo et al., 2022). Comorbidities and ASA scores significantly affected surgical outcomes. In conjunction with these findings, Sacko et al. advocated that age should not be considered a contraindication for surgery especially in patients with preoperative motor deficits, since surgery led to complete neurological recovery in 91.2% of the cases, and that most importantly, no death occurred, even in their cohort of patients aged 70 years and older (Sacko et al., 2009). However, it seems somehow surprising, but in the case of spinal meningioma, young age (<50 years) may indicate a risk factor for further surgery due to tumor recurrence (Cohen-Gadol et al., 2003; Kwee et al., 2020; Nakamura et al., 2012). That phenomenon might be attributable to the slow disease progression as well as to longer life expectancy in the younger age group. A concise overview of the related studies is provided in table 6.

Table 6: Overview of recent studies on spinal meningioma with a special focus on patients’ outcomes with respect to the patient’s age

n

Age, y

(mean)

Functional outcome

(% of improvement)

Recurrence rates

Reoperation

(Nakamura et al., 2012)

68

56

Not applicable

35.0%

(age < 50 y)

17.6%

(Cohen-Gadol et al., 2003)

40

34.5

82.0%

22.5%

22.5%

(Sacko et al., 2009)

102

74.6

91.2

1.0%

0.0%

(Kwee et al., 2020)

166

66

73.0%

7.2%

12.0%

(Hohenberger et al., 2020)

45

63

80.0%

None

0.0%

(Capo et al., 2022)

72

75

65.3%

None

None

A3. Discussion: We suggest to summarize results from other papers in one table.

Comment

We agree that a table presenting previous critical research is important to better understand the role of the patient’s age in disease management. Please also refer to our response to comment A2.2.

Reviewer 2 Report

The authors showed an interesting series of octogenarian patients surgically treated for spinal meningioma, with clear results and well discussed. Lacking in literature concerning series with this cut-off (≥80 years) makes the paper original, even if the cohort is small. Single institution assures homogeneous treatments and indications.

May the authors explain surgical indications in patients without neurological decline? Progressive growth lesion or spinal cord compression on MRI?

Line 56, I would add citations of references already present in the list + 1 recently published.

Morandi, X.; Haegelen, C.; Riffaud, L.; Amlashi, S.; Adn, M.; Brassier, G. Results in the Operative Treatment of Elderly Patients

with Spinal Meningiomas. Spine 2004, 29, 2191–2194.

Sacko, O.; Rabarijaona, M.; Loiseau, H. La Chirurgie Des Méningiomes Rachidiens Après 75 Ans. Neurochirurgie 2008, 54, 512–516.

Sacko O, Haegelen C, Mendes V, Brenner A, Sesay M, Brauge D, Lagarrigue J, Loiseau H, Roux F-E (2009) SPINAL MENINGI- 384 OMA SURGERY IN ELDERLY PATIENTS WITH PARAPLEGIA OR SEVERE PARAPARESIS: A MULTICENTER STUDY. 385 Neurosurgery 64:503–510. doi: 10.1227/01.NEU.0000338427.44471.1D

Capo, G.; Moiraghi, A.; Baro, V.; Tahhan, N.; Delaidelli, A.; Saladino, A.; Paun, L.; DiMeco, F.; Denaro, L.; Meling, T.R.; et al. Surgical Treatment of Spinal Meningiomas in the Elderly (75 Years): Which Factors Affect the Neurological Outcome? An International Multicentric Study of 72 Cases. Cancers 2022, 14, 4790. https://doi.org/10.3390/ cancers14194790

Line 71 Spine stability and exclusion criteria. Standing lateral radiograph should be the reference imaging for spine stability, more accurate of CT and MRI. I would not use the term « stability » but probably deformation, traumatic or severe degenerative changes. I would explain the reason of exclusion criteria. You fixed the patients with deformity/degenerative changes? Do you want to avoid bias in functional outcome?

Line 156 Authors should correct the phrase. N=5 , 16.7% refers only to acute heart failure. Add number of urinary infections please.  

Finally, I would have been more cautious in conclusions, considering the high rate of morality in-hospital.

Author Response

A1. May the authors explain surgical indications in patients without neurological decline? Progressive growth lesion or spinal cord compression on MRI?

Comment

We thank the reviewer for this insightful comment and agree that our inclusion criteria need further clarification.

The enrolled patients presented on admission with progressive neurological deficits; therefore, surgery was performed after careful monitoring the comorbid illnesses. No patient had an already radiologically confirmed spinal meningioma; therefore, progressive growth of lesions could not be evaluated.

To further clarify this point, we have added the following paragraph in the “Methods” section under the subsection “Study design and data collection”:

“Patients aged ³80 years with histologically confirmed SMs across the thoracic and lumbar spine and acute-onset neurological decline were consecutively enrolled. No patient had an already radiologically or histologically diagnosed spinal meningioma; thus, a disease progression or a progressive spinal cord compression on MRI due to the tumor mass could not be evaluated. Therefore, we did not include a cross-over.”

A2. Line 56, I would add citations of references already present in the list + 1 recently published.

 Comment

We have added the mentioned references.

A3. Line 71 Spine stability and exclusion criteria. Standing lateral radiograph should be the reference imaging for spine stability, more accurate of CT and MRI. I would not use the term « stability » but probably deformation, traumatic or severe degenerative changes. I would explain the reason of exclusion criteria. You fixed the patients with deformity/degenerative changes? Do you want to avoid bias in functional outcome?

Comment

To include the suggested changes, we have revised the “Methods” section as follows:

Spinal deformity, traumatic or degenerative changes were evaluated using computed tomography (CT) and magnetic resonance imaging (MRI).

The exclusion criteria have been revised as follows:

The exclusion criteria were as follows: age <80 years, concurrent intracranial or cervical pathology, complete loss of disc height, and bony deconstruction resulting in kyphosis or subluxation of the vertebral column, vertebral collapse of >50%, bone necrosis, spinal deformity, and traumatic or degenerative changes. This patient subset was excluded from the current study to avoid biases in the functional outcomes due to the need for different surgical techniques, e.g., spinal instrumentation and laminectomy for tumor resection. Such factors might have led to a heterogenous cohort, which would have significantly altered the functional outcomes and even the mortality rates due to extension of surgery.

A4. Line 156 Authors should correct the phrase. N=5 , 16.7% refers only to acute heart failure. Add number of urinary infections please.  

Comment

We sincerely apologize for omitting the urinary tract infection rates.

We have added the details as follows:

The most prevalent complications were urinary tract infections (n=5, 16.7%) and acute heart failure (n=5, 16.7%)

A5. Finally, I would have been more cautious in conclusions, considering the high rate of morality in-hospital.

Comment

We agree that we should be more cautious when interpreting our conclusions.

Therefore, we have revised the conclusions as follows:

“However, the surgical treatment and its related complications, including potential morbidity, should be discussed with patients and their relatives. Nevertheless, quality of life can be preserved or improved in octogenarians with prompt assessment and early surgery after acute onset.”
